# Incidence and Risk Factors for *Clostridioides difficile* Infections in Non-COVID and COVID-19 Patients: Experience from a Tertiary Care Hospital

**DOI:** 10.3390/microorganisms11020435

**Published:** 2023-02-08

**Authors:** Ljiljana Markovic-Denic, Vladimir Nikolic, Borislav Toskovic, Marija Brankovic, Bogdan Crnokrak, Viseslav Popadic, Aleksandra Radojevic, Dusan Radovanovic, Marija Zdravkovic

**Affiliations:** 1Faculty of Medicine, University of Belgrade, 11 000 Belgrade, Serbia; 2University Clinical Hospital Center Bezanijska Kosa, Faculty of Medicine, University of Belgrade, 11 000 Belgrade, Serbia

**Keywords:** healthcare-associated infections, *Clostridioides difficile*, COVID-19, incidence, risk factors

## Abstract

(1) Background: The aim of this study was to assess the incidence and the risk factors for healthcare-associated *Clostridioides difficile* infection (HA-CDI) in patients with COVID-19 and without this infection. (2) Methods: A single-center, prospective observational study was conducted at the University Clinical Hospital Center in Belgrade, Serbia, from January 2019 to December 2021. The entire hospital was a COVID-dedicated hospital for 12 months during the study period. The incidence density rates and risk factors for HA-CDI in patients with and without COVID-19 are presented. (3) Results: The incidence rates of HA-CDIs were three times higher in patients with COVID-19. The HA-CDI–COVID-patients were younger (69.9 ± 12.6 vs. 72.5 ± 11.6; *p* = 0.017), admitted from another hospital (20.5% vs. 2.9; *p* < 0.001), had antimicrobial therapy before CDI (99.1% vs. 91.3%, *p* < 0.001), received two or more antibiotics (*p* = 0.030) during a longer period (p = 0.035), received proton pump inhibitors (95.9% vs. 50.0%, *p* < 0.001) during a longer period (*p* = 0.012) and steroids (32.8% vs. 20.4%, *p* < 0.001). During the last month before their current hospitalization, a higher percentage of patients without COVID-19 disease were hospitalized in our hospital (*p* < 0.001). Independent predictors for HA-CDIs in patients with COVID-19 were admission from another hospital (*p* = 0.003), the length of antibiotic administration (0.020), and the use of steroids in therapy (*p* < 0.001). The HA-CDI predictors in the non-COVID patients were older age (*p* = 0.017), advanced-stage renal failure (*p* = 0.005), chemotherapy (*p* = 0.003), and a low albumin level (0.005). (4) Conclusion: Higher incidence rates of HAI-CDIs in COVID-19 patients did not occur due to reduced infection control precautions and hygiene measures but due to antibiotic therapy and therapy with other drugs used during the pandemic.

## 1. Introduction

Infections with *Clostridioides difficile* (CDIs) [1], formerly *Clostridium difficile*, are a leading cause of healthcare-associated infections (HAIs) and impact public health globally [2]. In the USA, the estimated incidence of community-associated *C. difficile* infection in 2017 was 70.4 per 100,000 population and the estimated national burden for healthcare-associated *C. difficile* infection (HA-CDI) was 73.3 (95% CI 68.9 to 77) [3]. According to the European Centre for Disease Prevention and Control [4], the crude incidence density of HA-CDI in 2016–2017 was 3.48 cases per 10,000 patient-days, with the highest rates in Lithuania (7.51 cases/10,000 patient-days), Poland (7.50 cases/10,000 patient-days), and Estonia (5.92 cases/10,000 patient-days). In the neighboring countries, Hungary had the highest rates (4.31 cases/10,000 patient-days) while Croatia had the smallest crude HA-CDI rates (2.50 cases/10,000 patient-days). Recently published results from a hospital in Serbia for the period 2011–2021 show that the lowest rate of *C. difficile* was 2.0 per 10,000 patient-days in 2018 and the highest was in 2019 (5.0 per 10,000 patient-days) [5]. In addition, with the growing CDIs threat in developed counties, there are important challenges in developing countries, as well [6]. Since 2019, the world has been faced with the COVID-19 pandemic. Patients hospitalized with COVID-19 had co-infections and superinfections with other hospital pathogens resulting in other hospital-acquired infections (HAI). Alterations in gut microbiota due to empiric antibiotic treatments and experimental antiviral and immunomodulatory drugs increased the risk for HA-CDI [7]. Moreover, conditions such as non-alcoholic fatty liver disease (NAFLD) can cause alterations in gut microbiota, thus increasing *Clostridioides difficile* colonization. On the other hand, NAFLD is a risk factor for obesity, which, besides other factors such as diabetes and hypertension, can increase the risk of severe COVID-19 [8].

In different countries, different approaches have been used for treating COVID-19 patients; entire buildings were converted into fully functioning hospitals, full hospitals or only parts of some hospitals were designated for treating COVID patients, or completely new COVID-19 hospitals were built [9,10,11]. In Serbia, different places for the healthcare of COVID-19 patients were also used. Before three new hospitals for COVID-19 patients were built, some hospitals were completely determined to be only for the treatment of COVID patients, while in others, only some wards were transformed into a COVID section.

The aim of this paper is to assess the incidence and the risk factors for HA-CDIs in patients with COVID-19 and without this infection.

## 2. Materials and Methods

### 2.1. Design and Study Setting

A single-center, prospective observational study was conducted at the University Clinical Hospital Center Bezanijska kosa, Belgrade, Serbia, from January 2019 to December 2021. After the first COVID-19 case in Serbia, during the first part of the epidemic in our county, only non-COVID-19 patients were admitted to our hospital. Since July 2020, the whole hospital has been transformed into a COVID-dedicated hospital, except in February and March 2021 and from May to September 2021. Therefore, we compared 24 months without COVID-19 patients (non-COVID period, when only non-COVID patients were hospitalized) and a 12-month COVID period (when only COVID-19 patients were treated in the whole hospital). During the COVID period, only patients diagnosed with COVID-19 were admitted to and hospitalized in our hospital. Our hospital is a 325-bed hospital with mainly multiple-occupancy rooms including 36 ICU beds. There are 15 single rooms and several double rooms in each department that can be used as single rooms for the isolation of patients with healthcare-associated infections, including CDI. During the COVID period, 35 respirators were placed in the ICU rooms. However, the total number of hospital beds was 260 due to the need for a greater number of isolation rooms and rooms with an oxygen supply.

### 2.2. Diagnosis and Definitions

The diagnosis of COVID-19 was based on clinical criteria and confirmed by a nasopharyngeal swab polymerase chain reaction (PCR). People with a new onset of at least three unformed stools in 24 h need *C. difficile* laboratory testing according to the World Health Organization (WHO) and the European Center for Disease Prevention and Control (ECDC) [12,13].

According to the guidelines published by professional societies, laboratory testing of CDI includes a two-step algorithm for optimal diagnostic accuracy [14,15,16]. First, testing with glutamate dehydrogenase (GDH) as a screening tool was performed. If the initial test was positive, additional testing with enzyme-linked immunosorbent assay (ELISA) was used for detecting the presence of toxigenic strains of *C difficile* [15]. We used CITEST^®^, CITEST DIAGNOSTICS Inc., Canada, and VIDAS^®^
*C. difficile* Toxin A&B (CDAB), bioMérieux SA. CDI was confirmed if both tests were positive. Discordant results, in the case of only GDH positive results, needed clinical consideration regarding treatment. Patients with only a first episode of CDI were included. Hospital-associated *C. difficile* infection (HA-CDI) was defined as CDI that occurred after 48 h of admission into a hospital or within four weeks of discharge from a healthcare facility [17].

### 2.3. Data Collection

Positive and negative CDI results were reported daily by clinical microbiologist to the hospital’s infection prevention and control (IPC) team. Data of new CDIs were analyzed by hospital epidemiologists using clinical data and data about current and previous hospitalization to confirm if these infections were HA-CDI. Community-acquired CDIs were excluded. Patient demographics data, hematological data, and biochemical markers were collected from the hospital patient administration system connected with the local laboratory information system. We compared age, gender, comorbidities, from whence they were admitted to the hospital, therapy before the diagnosis of CDI, clinical manifestation and CDI therapy, laboratory findings and outcome in patients without COVID-19, and patients with laboratory-confirmed SARS-CoV-2 infections hospitalized in our hospital during two study periods.

### 2.4. Ethical Consideration

The study protocol was approved by the University Clinical Hospital Center Bezanijska Kosa ethics committee (protocol number 9941, date 25 December 2018), according to the principles of the Declaration of Helsinki of 1975, as revised in 2008.

### 2.5. Statistical Analysis

The local IPC team entered all of the data into a previously prepared database. The results were expressed as the mean ± SD or the proportion of total number of included patients. The incidence density rate was calculated as the number of new HA-CDI cases divided by the number of patient-days of hospitalization expressed per 1000 patient-days at risk, with a 95% CI. Changes in rates over time were presented graphically. The incidence rate ratios (IRRs) with a 95% CI were calculated by comparing COVID-19 patients and non-COVID patients. The normality of distribution was verified with the Shapiro–Wilk test. The chi-square test was used to compare the categorical data. The Student’s *t*-test or Mann–Whitney U test were used to compare continuous variables, depending on the normality of the distribution. Univariable and multivariable logistic regression analyses were performed to identify predictors of CDI in patients with COVID-19. Variables that showed significant univariate logistic regression at a *p*-value less than 0.05 were included in the multivariate logistic regression. Odds ratios (OR) with 95% confidence intervals (CIs) were computed and the Hosmer–Lemeshow goodness-of-fit test was performed to assess the overall model fit. The statistical analysis was performed using SPSS version 17.0 software (SPSS Inc., Chicago, IL, USA).

## 3. Results

Out of 547 *C. difficile* cases enrolled during the three-year study period, 341 (62.3%) were identified in SARS-CoV-2-infected patients. The incidence density of HO-CDIs per 1000 patients-days was 1.33 in the non-COVID period and 4.53 in the COVID period. The IRR was 3.41 (95% CI 2.86 to 4.08) when compared the COVID-19 patients and non-COVID patients. The incidence density of HO-CDIs was 1.06 per 1000 patient-days in the pre-pandemic period, during 2019. During the pandemic, the rate of HA-CDI in non-COVID patients was 0.6 per 1000 patients-days in 2020, but 3.11 per 1000 patients-days in 2021. During the two pandemic years, when the hospital was a COVID-19-dedicated hospital, the rates for COVID-positive patients were 2.5 (in 2020) and 6.5 per 1000 patients-days in 2021 (Figure 1).

The demographic and clinical characteristics of included patients are presented in Table 1. The patients with SARS-CoV-2 infection were statistically younger than those without (69.9 vs. 72.5 years). About a quarter of them were transferred to our hospital from another hospital or nursing home (*p* < 0.001). During the last month before their current hospitalization, the SARS-CoV-2 patients were frequently hospitalized in another hospital (*p* = 0.018), while a higher percentage of patients without COVID-19 were previously hospitalized in our hospital (*p* < 0.001), without a difference in the length of their hospital stay. There was no significant difference in the length of hospitalization and the outcome between COVID-19 patients and non-COVID patients.

The statistically higher percentage of patients without COVID-19 disease had comorbidities such as coronary heart disease, chronic obstructive lung disease, renal failure, and malignancy. No statistically significant difference was shown between the non-COVID and COVID-19 patients regarding endocrine diseases (*p* = 0.760) (Table 2). Regarding type 2 DM, there was no difference in the types of therapy (oral, insulin, or combined) between the CDI patients with and without COVID-19 (*p* = 0.797). Although all of the CDI patients had higher first fasting blood levels a day after admission, the differences were not significant among the COVID-19 and non-COVID patients: median 8.4 (3.0–38.4) mmol/l for non-COVID patients and 10.1 (1.6–35.0) mmol/l for COVID-19 patients (*p* = 0.382 (data not shown in Table 2).

Overall, 99.1% of the SARS-CoV-2-infected patients were exposed to antibiotics up to one month before the diagnosis of *C. difficile* infection versus 91.3% of patients without COVID-19 disease (*p* < 0.001). In the COVID-19 patients, several different antibiotics were used in therapy. Two or more antibiotics in therapy were more frequently used in the COVID-19 patients as well as the duration of antibiotic therapy, with the use of three antibiotics also being longer in these patients (*p* = 0.030 and *p* = 0.035 respectively). Exposition to chemotherapy, H2-receptor antagonists, and other types of surgery (except abdominal surgery) was significantly higher in patients without COVID-19 disease. In addition, surgery during ongoing hospitalization was more frequent in these patients. On the other hand, SARS-CoV-2-infected patients were significantly more exposed to proton pump inhibitors (*p* < 0.001), probiotics (*p* < 0.001), benzodiazepines (*p* = 0.002), and steroids (*p* < 0.001). The characteristics of the used therapy before the diagnosis of *C. difficile* infection are shown in Table 3.

The typical clinical manifestations of *C. difficile* infection were more frequently presented in the patients without COVID-19 disease (Table 4). Statistically, significantly more patients without COVID-19 had high creatinine levels (38.6 vs. 19.5), low albumin levels (79.7 vs. 58.1), and high CRP levels (96.4 vs. 83.3) versus the SARS-CoV-2-positive patients. The most common treatment for *C. difficile* infection was metronidazole while 87.6% of patients received both metronidazole and vancomycin. Therapy with vancomycin was significantly more often used in patients without COVID-19 disease but patients with SARS-CoV-2 infection had prolonged treatment after discharge in a higher percentage (Table 4).

The factors associated with *C. difficile* infection in COVID-19 patients identified through multivariate logistic regression were age, from whence the patients were admitted to the hospital, advanced stage of renal failure, exposition to chemotherapy, length of administration of one antibiotic, steroid use, and low albumin levels (Table 5).

## 4. Discussion

Our study revealed that the incidence density rate was three times higher when the hospital was a COVID-dedicated hospital, i.e., when only COVID-19-positive patients were hospitalized in it, than during the period when it was a non-COVID hospital, before and during the COVID-19 pandemic. The incidence of CDIs increased in the middle of the first decade of the 21st century due to new, highly virulent *C. difficile* strains such as ribotype (RT) 027 [18]. Improvement in the rational use of antibiotics, along with other infection prevention and control measures, as the essential factors in the prevention of CDI became the national priority in many countries and the subject of organized surveillance [17,19]. Then, it was observed that HA-CDIs decreased by 36% in the USA during 2011–2017 [3] due to efforts in implementing preventive measures, especially antibiotic stewardship, which reduced fluoroquinolone use in their hospitals [20,21]. Similar trends were observed in EU countries [4]. In the pre-pandemic period, the incidence rate obtained in our study was similar to those in some EU countries, Italy for example, but higher than in other EU countries with high HA-CDI incidence rates [4,22].

However, when the COVID-19 pandemic began, the observed data were disparate. HA-CDI rates did not change in the US hospitals [23] or even decreased in some EU countries [24,25]. In contrast, the incidence rates of HA-CDIs increased in hospitals participating in the Canadian Nosocomial Infection Surveillance Program [26], after a period of steady decline before the pandemic. This can be explained by altered or even weakened infection prevention and control practices due to pandemic surge pressures and increased antibiotic use. According to the meta-analysis of articles published up to June 2020, every three out of four COVID-19 patients worldwide received antibiotics [27].

During the COVID pandemic period, our results are in concordance with the results from other studies in which higher HA-CDI rates were observed in COVID-19 patients [27,28]. Inappropriate antibiotic prescription in COVID units of primary healthcare centers in our country for patients with middle COVID-19 infections has led to the tremendous use of antibiotics, a well-known risk factor for CDI. During the first half of the first pandemic year, the proportion of COVID-19 patients who received antibiotics varied from 63.1% in Europe, 64.8% in North America, and 76.2% in China, to even 87.5% in other parts of East–Southeast Asia [27].

In addition, many patients in the non-COVID hospital received antibiotics for the treatment of underlying diseases, which continued with even higher doses in the COVID hospital when they were transferred to it as they were positive for the SARS-CoV-2 virus. According to our results, a significantly higher proportion of COVID-19 patients with HA-CDIs were previously treated in other hospitals from whence they were transferred to our COVID hospital. Admission from another hospital increased the risk for HA-CDI by 33 times in COVID-19 patients. Our results show that significantly more COVID-19-positive patients received antibiotics before being hospitalized in our hospital, by as much as 99%. The length of administration of an antibiotic was an independent risk factor for HA-CDIs in COVID-19 patients obtained in the multivariable logistic regression analysis.

It was observed in our country that 34% to more than half of patients used antibiotics before hospital admission and more than three-quarters used antibiotics during hospitalization in COVID hospitals [29,30]. Broad-spectrum antimicrobial drugs can change the normal intestinal microbiota resulting in CDI appearance [31].

In our study, steroid use was an independent risk factor for HA-CDI in COVID-19 patients. Similar results have been shown in a study conducted in other COVID and non-COVID hospitals in our country [32]. These drugs were part of many versions of protocols for the treatment of COVID-19 patients due to their anti-inflammatory role. However, there were negative attitudes about their use in non-oxygen requiring COVID-19 patients [33]. Moreover, in the overview of the systematic review about their effect in the treatment of COVID-19 patients, no firm conclusion was found about their role in reducing disease progression and mortality [34]. The studies conducted in the pre-pandemic period showed mixed results regarding steroids’ effect on CDI development [35,36,37].

The results of the multivariable logistic regression in our study showed that hypoalbuminemia was statistically significantly more common in non-COVID-CDI patients. It is known that hypoalbuminemia is linked to poor patient health and can be associated with the development of different HAIs. Low serum albumin levels indicate a higher risk of acquiring CDIs [38]. The protective effects of serum albumin on *C. difficile*-induced host cell damage has been proven in vitro and in vivo [39].

We found that COVID-19 patients who received proton pump inhibitors in their therapy had a 23-times higher risk of HA-CDIs than non-COVID patients. These acid-suppressive medications are a well-known risk factor for CDIs because they facilitate vegetative *C. difficile* survival and growth leading to CDI occurrence [40]. However, the role of proton pump inhibitors as the risk factor for CDIs is still controversial [41]. The analysis of the results of a European, multi-center, bi-annual point prevalence study of *C. difficile* infection in hospitalized patients with diarrhea (EUCLID), which included 59 hospitals in seven countries across Europe, did not show a significant association between PPIs and CDI. However, in other groups of CDI cases and controls from one university hospital in Germany, this association was confirmed [42].

Regarding the demographic characteristics of our patients, gender differences were not observed between the patients in the two study groups. The mean age of patients with HA-CDI was 71 years, while non-COVID-19 patients were older. The results of other studies have shown that there were no differences in the CDI patients’ age before and during the COVID-19 pandemic [28] or that the COVID-19 patients were older [25]. A higher frequency of underlying diseases and comorbidities such as coronary heart disease, chronic pulmonary disease, advanced-stage renal failure, and cancer in non-COVID patients can probably be explained by their older age.

Malignancy was found to be more frequent in patients without COVID-19 which was also shown in Spain [25]. In contrast to our findings, a study conducted also in our county showed that chronic renal failure, malignancy, and chronic obstructive lung disease are more frequent in HA-CDI–COVID-19 patients than in HA-CDI patients without COVID-19 [32]. A possible explanation for these differences is that this study was conducted in two separate hospitals. In contrast, our study was carried out in the same hospital, which was transformed several times into a hospital for treating only COVID-19 patients during the first two years of the COVID-19 pandemic. Before the pandemic, the hospital treated many patients with various diseases, especially those with chronic heart disease, chronic obstructive lung disease, advanced-stage renal failure, and cancer. There were patients who require repeated hospitalization due to such diseases. These patients are residents of the city area who gravitate towards our hospital according to spatial distribution and accessibility of tertiary care hospitals. During the COVID-19 pandemic, when patients became COVID-19-positive in other non-COVID hospitals, they were transferred to our COVID-dedicated hospital.

Regarding the clinical manifestation and laboratory findings of the HA-CDI patients in our study, it was found that typical symptoms of CDIs, such as abdominal pain, stomach cramps, and nausea or vomiting, were more frequent in HA-CDI patients without COVID-19 disease. Although those gastrointestinal symptoms are recognized as frequent in COVID-19 patients [43,44], we noticed a higher proportion of these symptoms in HA-CDI–non-COVID patients, contrary to the finding in the literature [7]. Preoccupation with pulmonary symptoms in COVID-19 patients may have led to the neglect of these symptoms of CDI infection. However, equal attention in both groups of patients was focused on the occurrence of diarrhea, so we believe there was no underestimation of the number of HA-CDI in COVID-19 patients.

Besides the low levels of serum albumin revealed by multivariable logistic regression, higher creatinine levels and CRP were observed in a higher proportion of HA-CDI–non-COVID patients by univariate analysis. Creatinine and CRP were singled out as independent predictors of a severe form of CDI and were attributed to mortality in patients with CDI before the start of the pandemic [43]. Although a negative impact of associated CDI and COVID-19 on patient outcomes was found [45], we did not find a difference in outcomes between COVID and non-–COVID HA-CDI patients. Our results are in concordance with those from Spain [25]. The fatal outcome of COVID-19-positive patients was mainly due to the severity of this disease in our patients. Namely, 86% of COVID-19 patients who died had a CT score at admission higher than seven, which indicates the severity of the disease and a potentially bad disease outcome [46,47].

One of the main limitations of this study is the single-center design. On the other hand, it is also an advantage of the study because the non-COVID and COVID-19 patients were treated in the same hospital. Thus, there was no difference between the hospital staff, patient care, and the conditions in which the patients were treated. The second advantage of this study is that the same IPC team monitored HAI and CDI in both of the study periods. Another limitation is the absence of data on ribotyping. This is not routinely performed in our country, but only in specialist research. According to the results of such research, *Clostridium difficile* ribotype 027 was the most prevalent ribotype before the pandemic [48].

## 5. Conclusions

We believe that the increase in HAI-CDI in COVID-19 patients did not occur due to changes in infection control precautions and hygiene measures, given that the same infection control team performed continuous surveillance and organized preventive measures for all of the patients in both study periods. On the contrary, even stricter measures were applied during the COVID-19 epidemic. It is more likely that other risk factors, such as previous hospitalization in another hospital, antibiotic use, the length of antibiotic administration, and steroid use, contributed to the higher incidence of HA-CDI in COVID-19 patients.

## Figures and Tables

**Figure 1 microorganisms-11-00435-f001:**
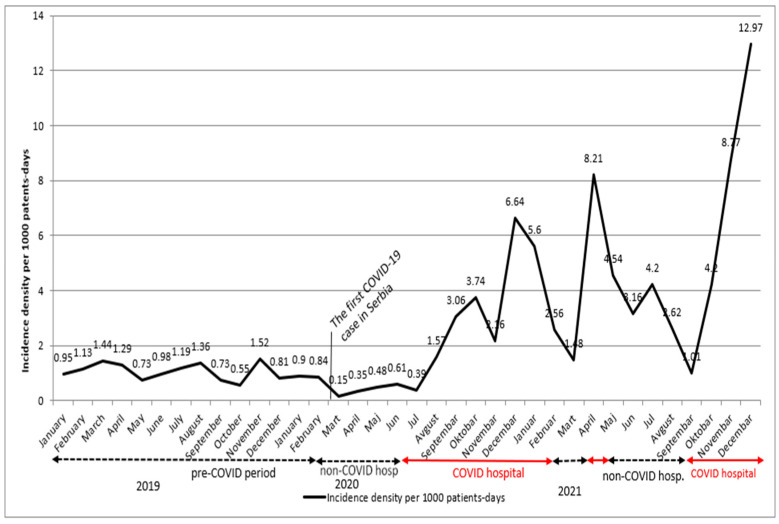
HA-CDI incidence density throughout the three-year period.

**Table 1 microorganisms-11-00435-t001:** Demographic and clinical characteristics of the study population.

	Patients without COVID-19 Disease	COVID-19 Patients	Total	*p* Value
	N (%)			
Total	206 (37.7)	341 (62.3)	547 (100.0)	
Gender:				0.546
Male	96 (46.6)	168 (49.3)	264 (48.3)	
Female	110 (53.4)	173 (50.7)	283 (51.7)	
Age—years (mean ± SD)	72.5 ± 11.6	69.9 ± 12.6	70.9 ± 12.3	**0.017**
BMI (mean ± SD)	27.6 ± 6.9	27.5 ± 4.7	27.5 ± 5.5	0.900
**Admitted to hospital:**				
From home	195 (94.7)	260 (76.2)	455 (83.2)	**<0.001**
From another hospital	6 (2.9)	70 (20.5)	76 (13.9)	
From nursing home	5 (2.4)	11 (3.2)	16 (2.9)	
Prior hospital stay in the last month	73 (35.4)	97 (28.4)	170 (31.1)	0.087
Prior hospital stay in the same hospital	43 (20.9)	16 (4.7)	59 (10.8)	**<0.001**
Length of prior hospital stay in the same hospital—days (mean ± SD)	11.7 ± 8.8	10.2 ± 5.6	11.3 ± 8.0	0.537
Prior hospital stay in the another hospital	32 (15.5)	82 (24.0)	114 (20.8)	**0.018**
Length of prior hospital stay in another hospital—days (mean ± SD)	12.5 ± 9.0	9.1 ± 5.7	10.0 ± 6.8	0.091
**Outcome**				
Dismissed to go home	155 (75.2)	270 (79.2)	425 (77.7)	0.387
Moved to another hospital	0 (0.0)	1 (0.3)	1 (0.2)	
Fatal outcome	51 (24.8)	70 (20.5)	121 (22.1)	
Length of hospitalization—days (mean ± SD)	21.2 ± 12.8	20.1 ± 11.2	20.5 ± 11.8	0.272

SD—standard deviation.

**Table 2 microorganisms-11-00435-t002:** Comorbidities of the patients included in the study.

	Without COVID-19 Disease n (%)	COVID-19 Patients n (%)	Total n (%)	*p* Value
Hypertension	148 (71.8)	226 (66.3)	374 (68.4)	0.175
Coronary heart disease	64 (31.1)	56 (16.4)	120 (21.9)	**<0.001**
Chronic obstructive lung disease	28 (13.7)	14 (4.1)	42 (7.7)	**<0.001**
Advanced stage of renal failure	19 (9.2)	9 (2.6)	28 (5.1)	**0.001**
Malignancy	52 (25.2)	21 (6.2)	73 (13.3)	**<0.001**
Inflammatory bowel disease	3 (1.5)	2 (0.6)	5 (0.9)	0.300
Stroke	8 (3.9)	11 (3.2)	19 (3.5)	0.684
Chronic liver disease (compensated cirrhosis)	5 (2.4)	2 (0.6)	7 (1.3)	0.063
Endocrine diseases	63 (30.6)	105 (30.8)	168 (30.7)	0.959
Diabetes mellitus type 2	56 (27.2)	78 (22.9)	134 (24.5)	0.256
Hyperthyroidism	2 (1.0)	3 (0.9)	5 (0.9)	
Hypothyroidism	11 (5.3)	24 (7.0)	35 (6.4)	0.760
Hyperlipidemia	0 (0.0)	1 (0.3)	1 (0.2)	
Hypercholesterolemia	0 (0.0)	1 (0.3)	1 (0.2)	

**Table 3 microorganisms-11-00435-t003:** Characteristics of therapy before the diagnosis of *C. difficile* infection.

	Without COVID-19 Disease n (%)	COVID-19 Patients n (%)	Total n (%)	*p* Value
Antibiotic therapy up to one month before the diagnosis of *C. difficile* infection	188 (91.3)	338 (99.1)	526 (96.2)	**<0.001**
Number of antibiotics used in therapy				
One	64 (38.1)	87 (25.9)	151 (30.0)	0.030
Two	55 (32.7)	118 (35.1)	173 (34.3)	
Three	30 (17.9)	75 (22.3)	105 (20.8)	
Four and more	19 (11.3)	56 (16.7)	75 (14.9)	
Length of administration of one antibiotic—days (mean ± SD)	5.1 ± 3.6	5.6 ± 3.3	5.4 ± 3.5	0.364
Length of administration of two antibiotics—days (mean ± SD)	10.0 ± 7.2	11.0 ± 5.5	10.7 ± 6.1	0.288
Length of administration of three antibiotics—days (mean ± SD)	13.8 ± 9.5	17.7 ± 7.8	16.6 ± 8.5	**0.035**
Length of administration of four and more antibiotics—days (mean ± SD)	28.0 ± 18.8	27.3 ± 14.8	27.5 ± 15.8	0.866
Chemotherapy	23 (11.2)	6 (1.8)	29 (5.3)	**<0.001**
H2-receptor antagonists	34 (16.5)	0 (0.0)	34 (6.2)	**<0.001**
Proton pump inhibitors	103 (50.0)	327 (95.9)	430 (78.6)	**<0.001**
Proton pump inhibitors therapy duration—days (mean ± SD)	8.6 ± 7.5	11.2 ± 8.0	10.7 ± 8.0	**0.012**
Probiotics	123 (60.0)	303 (88.9)	426 (78.0)	**<0.001**
Probiotics therapy duration—days (mean ± SD)	10.1 ± 9.0	11.2 ± 8.0	10.9 ± 8.2	0.251
Statins	23 (11.2)	30 (8.8)	53 (9.7)	0.364
Steroids	85 (41.3)	315 (92.4)	400 (73.1)	**<0.001**
Abdominal surgery in the last month	8 (3.9)	6 (1.8)	14 (2.6)	0.127
Other surgery in the last month	15 (7.3)	4 (1.2)	19 (3.5)	**<0.001**
Surgery during ongoing hospitalization	19 (9.2)	3 (0.9)	22 (4.0)	**<0.001**

**Table 4 microorganisms-11-00435-t004:** Clinical manifestation and laboratory analyses of *C. difficile* infection for non-COVID and COVID-19 patients.

	Without COVID-19 Disease n (%)	COVID-19 Patients n (%)	Total n (%)	*p* Value
Mean days from admission to laboratory confirmation of *C. difficile* infection	10.5 ± 8.6	11.0 ± 7.9	10.8 ± 8.2	0.547
**Clinical manifestation**				
Abdominal pain	22 (10.7)	13 (3.8)	35 (6.4)	**0.001**
Stomach cramps	10 (4.9)	4 (1.2)	14 (2.6)	**0.008**
Nausea/vomiting	15 (7.3)	7 (2.1)	22 (4.0)	**0.003**
**Laboratory analyses**				
High leukocytes level (>10 × 10^9^/L)	127 (62.6)	198 (58.1)	325 (59.7)	0.301
High neutrophils level (>7.5 × 10^9^/L)	126 (62.1)	215 (63.4)	341 (62.9)	0.752
High lymphocytes level (>4 × 10^9^/L)	5 (2.5)	6 (1.8)	11 (2.0)	0.573
High creatinine level (>106 umol/L)	78 (38.6)	66 (19.5)	144 (26.7)	**<0.001**
High LDH level (>270 U/L)	72 (54.5)	187 (61.3)	259 (59.3)	0.186
Low albumin level (<35 g/L)	118 (79.7)	161 (58.1)	279 (65.6)	**<0.001**
High CRP level (>5 mg/L)	188 (96.4)	284 (83.3)	472 (88.1)	**<0.001**
Sedimentation rate (mean ± SD)	41.86 ± 20.86	33.90 ± 28.57	36.56 ± 26.35	0.262

SD—standard deviation; LDH—actate dehydrogenase; CRP—C-reactive protein.

**Table 5 microorganisms-11-00435-t005:** Univariate and multivariate logistic regression for identifying factors associated with *C. difficile* infection in patients with COVID-19 disease.

	Univariate Logistic Regression		Multivariate Logistic Regression	
	OR (95% CI)	*p* Value	OR (95% CI)	*p* Value
Age	**0.98 (0.97–0.99)**	**0.018**	**0.95 (0.91–0.99)**	**0.017**
Gender (female vs. male)	0.90 (0.64–1.27)	0.546		
BMI	0.99 (0.92–1.07)	0.898		
**Admitted to hospital:**				
From home	ref.		ref.	
From another hospital	0.61 (0.21–1.77)	0.360	**33.04 (3.40–320.83)**	**0.003**
From nursing home	**5.30 (1.38–20.38)**	**0.015**	1.08 (0.15–7.57)	0.939
Prior hospital stay in the last month	0.72 (0.50–1.05)	0.087		
Prior hospital stay in the same hospital in the last month	**0.19 (0.10–0.34)**	**<0.001**		
Prior hospital stay in the another hospital	**1.72 (1.10–2.70)**	**0.018**		
Abdominal surgery in the last month	0.44 (0.15–1.30)	0.137		
Other surgery in the last month	**0.15 (0.05–0.46)**	**0.001**		
ICU hospitalization	0.89 (0.55–1.43)	0.628		
Surgery during ongoing hospitalization	**0.09 (0.03–0.30)**	**<0.001**		
Hypertension	0.77 (0.53–1.12)	0.175		
Coronary heart disease	**0.44 (0.29–0.66)**	**<0.001**	0.41 (0.16–1.08)	0.072
Chronic obstructive pulmonary disease	**0.27 (0.14–0.53)**	**<0.001**		
Advanced stage of renal failure	**0.27 (0.12–0.60)**	**0.001**	**0.10 (0.02–0.50)**	**0.005**
Cancer	**0.19 (0.11–0.33)**	**<0.001**		
Inflammatory bowel disease	0.40 (0.07–2.41)	0.317		
Stroke	0.82 (0.33–2.09)	0.684		
Chronic liver disease	0.24 (0.05–1.23)	0.087		
Endocrine diseases	1.01 (0.69–1.47)	0.959		
Diabetes mellitus type 2	0.79 (0.53–1.18)	0.257		
Chemotherapy	**0.14 (0.06–0.36)**	**<0.001**	**0.06 (0.01–0.38)**	**0.003**
Antibiotic therapy up to one month before the diagnosis of *C. difficile* infection	**10.79 (3.14–37.10)**	**<0.001**		
Number of antibiotics used in therapy				
One	ref.			
Two	**1.58 (1.002–2.49)**	**0.049**		
Three	**1.94 (1.08–3.13)**	**0.025**		
Four and more	**2.17 (1.17–4.00)**	**0.013**		
Length of administration of one antibiotic	**1.03 (1.01–1.05)**	**0.005**	**1.06 (1.01–1.11)**	**0.020**
Proton pump inhibitors	**23.36 (12.81–42.58)**	**<0.001**		
Proton pump inhibitors therapy duration—days	**1.05 (1.01–1.09)**	**0.013**		
Probiotics	**5.32 (3.43–8.24)**	**<0.001**		
Probiotics therapy duration—days	1.02 (0.99–1.05)	0.252		
Statins	0.77 (0.43–1.36)	0.365		
Steroids	**17.25 (10.60–28.06)**	**<0.001**	**19.69 (7.70–50.34)**	**<0.001**
High creatinine level	**0.39 (0.26–0.57)**	**<0.001**		
Low albumin level	**0.35 (0.22–0.56)**	**<0.001**	**0.41 (0.16–0.99)**	**0.050**
High LDH level	1.32 (0.87–2.00)	0.187		
High CRP level	**0.19 (0.08–0.41)**	**<0.001**	0.17 (0.03–1.02)	0.052

ICU—intensive care unit; LDH–lactate dehydrogenase; CRP—C-reactive protein.

## Data Availability

The data presented in this study are available on request from the corresponding author. The data are not publicly available due to privacy.

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
