# Peer review of "Incidence and Risk Factors for Clostridioides difficile Infections in Non-COVID and COVID-19 Patients: Experience from a Tertiary Care Hospital"

_microorganisms, 2023, doi:10.3390/microorganisms11020435_

Round 1

Reviewer 1 Report

This is a well-designed study that compares the rates of CDI in patients with and without COVID 19. In my opinion results are relevant and novel. What is the most striking for me is the rate of antibiotic use in this Serbian hospital for COVID 19 patients, which is much lower that what we use in North America. 

I have several points for authors to add/clarify in order to further improve the paper: 

1. Line 44- since this study is from Serbia- do you have information to add about Serbia and how it compares to other European countries?

2. Line 50- patients with NAFLD and obesity have higher risk for CDI development and are more likely to be hospitalized for COVID so this is another link and should be mentioned. See the following : Adenote A, Dumic I, Madrid C, Barusya C, Nordstrom CW, Rueda Prada L. NAFLD and Infection, a Nuanced Relationship. Can J Gastroenterol Hepatol. 2021 Apr 15;2021:5556354. doi: 10.1155/2021/5556354. PMID: 33977096; PMCID: PMC8087474.

3. Please state in methods what was the hospital capacity and how many of the total beds were ICU designated? Additionally- are the rooms single occupancy or more patients shared the same room?

4. Line 94- IPC team. I suspect this is infection prevention control team. Please expend abbreviation. 

5. Line 143- " during the last month"? Last month of the study, perhaps? 

6. Table 1- prior hospital stay in the same hospital was statistically significant. However, this was not mentioned in the results section of the abstract so please correct that. Furthermore- the reasons why this might be the case should be elaborated in discussion. 

7. Mortality- in table 1: was this due to COVID 19 and its complication or any of these were due to CDI? Please specify further.

8. Table 2- chronic liver disease and endocrine disease are very nonspecific. For example, there is a difference regarding the risk for COVID 19 and CDI in patients who have highly uncontrolled type 2 DM and patients with well controlled hypothyroidism- yet both of these groups seem to be placed together. The same criticism is regarding chronic liver disease- there is a difference regarding patient with compensated and decompensated cirrhosis. 

9. Line 156- " more different antibiotics" is unclear. Please rewrite

10. Line 164- please remove hormone- steroids is sufficient

11. 

Author Response

Reviewer 1#

This is a well-designed study that compares the rates of CDI in patients with and without COVID 19. In my opinion results are relevant and novel. What is the most striking for me is the rate of antibiotic use in this Serbian hospital for COVID 19 patients, which is much lower that what we use in North America. 

I have several points for authors to add/clarify in order to further improve the paper: 

  1. Line 44- since this study is from Serbia- do you have information to add about Serbia and how it compares to other European countries?

Response: We thank the reviewer for this valuable comment. We added the date from another hospital in Belgrade and the date on C. difficile rates in the  neighboring countries.

  1. Line 50- patients with NAFLD and obesity have higher risk for CDI development and are more likely to be hospitalized for COVID so this is another link and should be mentioned. See the following : Adenote A, Dumic I, Madrid C, Barusya C, Nordstrom CW, Rueda Prada L. NAFLD and Infection, a Nuanced Relationship. Can J Gastroenterol Hepatol. 2021 Apr 15;2021:5556354. doi: 10.1155/2021/5556354. PMID: 33977096; PMCID: PMC8087474.

Response: We thank the reviewer for this important suggestion. We have now cited this reference and explained the link between NAFLD and obesity with CDI and COVID-19.

  1. Please state in methods what was the hospital capacity and how many of the total beds were ICU designated? Additionally- are the rooms single occupancy or more patients shared the same room?

Response: Thank you very much for this constructive input. We added in Methods an explanation about the structure of our hospital and the number of beds, including ICU beds, during the non-COVID and COVID periods.

  1. Line 94- IPC team. I suspect this is infection prevention control team. Please expend abbreviation. We

Response: We added the full name to the abbreviation.

  1. Line 143- " during the last month"? Last month of the study, perhaps? 

Response: Thank you for this suggestion. We added that the previous hospitalization refers to the last month before the current hospitalization.

  1. Table 1- prior hospital stay in the same hospital was statistically significant. However, this was not mentioned in the results section of the abstract so please correct that. Furthermore- the reasons why this might be the case should be elaborated in discussion. 

Response: Thank you very much for this constructive input. We added this result in the abstract. We also added one paragraph in the Discussion and explained the reasons for the re-hospitalization of patients in our hospital before the pandemic. During the pandemic, when our hospital was a COVID-dedicated hospital, the patients were hospitalized in another hospital before they became COVID-positive.

  1. Mortality- in table 1: was this due to COVID 19 and its complication or any of these were due to CDI? Please specify further.

Response: We appreciate this comment. It is complicated to prove the attribution of CDI to COVID-19 patients’ mortality. We did not find a difference in outcomes between COVID and non-COVID CDI patients. The fatal outcome of COVID-19-positive patients was mainly due to the severity of this disease. Namely, 86% of COVID-19 patients who died had a CT score at admission higher than seven, which indicates the severity of the disease and potentially bad disease outcome. We added this explanation in the Discussion.

  1. Table 2- chronic liver disease and endocrine disease are very nonspecific. For example, there is a difference regarding the risk for COVID 19 and CDI in patients who have highly uncontrolled type 2 DM and patients with well controlled hypothyroidism- yet both of these groups seem to be placed together. The same criticism is regarding chronic liver disease- there is a difference regarding patient with compensated and decompensated cirrhosis. 

Response: We thank the reviewer for this valuable comment. All patients had compensated cirrhosis, and we changed it in Table 2. Regarding diabetes mellitus, glycated hemoglobin (Hb1Ac) test is not performed on all patients admitted to the hospital. Therefore, we have only anamnesis data about diabetes therapy and data on routine blood glucose measurement on admission. There was no difference regarding the type of DM therapy (oral, insulin, or combined) between CDI patients with and without COVID-19 (p=0.797). Although all CDI patients had higher first fasting blood levels a day after admission, the differences were not significant among COVID and non-COVID patients: median 8.4 (3.0-38.4) mmol/l for non-COVID and 10.1 (1.6-35.0) mmol/l for COVID-19 patients (p=0.382, Mann-Whitney U test). We explained it in the Results.

Regarding endocrine diseases other than type 2 DM we added in table 2 an addition. For hyperthyroidism and hypothyroidism, we only had anamnestic data that all patients were on chronic therapy without any symptoms related to these diseases. There was one patient, each with hypocholesterolemia and hyperlipidemia (both with COVID-19), as well on chronic therapy. No statistically significant difference was shown between non-COVID and COVID-19 patients regarding endocrine diseases (p=0.760).

  1. Line 156- " more different antibiotics" is unclear. Please rewrite

Response: Thank you for this suggestion. We changed this sentence as follows:  In COVID-19 patients, several different antibiotics were used in therapy.

  1. Line 164- please remove hormone- steroids is sufficient.

Response: Thank you, we changed it.

Reviewer 2 Report

dear authors

Introduction

At several points in the work the scientific names are not written in italics

Materials and methods

Add ethics committee approval data, protocol number, committee name

Author Response

Reviewer 2#

Introdução

Em vários pontos do trabalho os nomes científicos não são escritos em itálico

Introduction

At several points in the work the scientific names are not written in italics

Response: We thank the reviewer to this comment. We changed all names of bacteria to italics.

Materiais e Métodos

Adicionar dados de aprovação do comitê de ética, número de protocolo, nome do comitê

Materials and methods

Add ethics committee approval data, protocol number, committee name

Response: We added the full name of the Ethics committee, and added the date and protocol number.

Round 2

Reviewer 1 Report

I would like to thank the authors for the detailed responses to my comments. My concerns have been addressed, and I have no further remarks. I recommend acceptance of the paper in the current form. Congratulations. 

Reviewer 2 Report

No comments